# Screen-Printed Glucose Sensors Modified with Cellulose Nanocrystals (CNCs) for Cell Culture Monitoring

**DOI:** 10.3390/bios10090125

**Published:** 2020-09-13

**Authors:** Ye Tang, Konstantinos Petropoulos, Felix Kurth, Hui Gao, Davide Migliorelli, Olivier Guenat, Silvia Generelli

**Affiliations:** 1Swiss Center for Electronics and Microtechnology (CSEM, Landquart), Bahnhofstrasse 1, 7302 Landquart, Switzerland; ye.tang@csem.ch (Y.T.); konstantinos.petropoulos@csem.ch (K.P.); felix.kurth@csem.ch (F.K.); hui.chai-gao@csem.ch (H.G.); davide.migliorelli@csem.ch (D.M.); 2Organs-on-Chip Technologies, ARTORG Center for Biomedical Engineering Research, University of Bern, Murtenstrasse 50, 3008 Bern, Switzerland; olivier.guenat@artorg.unibe.ch

**Keywords:** cellulose nanocrystals, amperometric glucose sensor, cell culture monitoring, screen-printed electrodes, long-term stability

## Abstract

Glucose sensors are potentially useful tools for monitoring the glucose concentration in cell culture medium. Here, we present a new, low-cost, and reproducible sensor based on a cellulose-based material, 2,2,6,6-tetramethylpiperidine-1-oxyl (TEMPO) oxidized-cellulose nanocrystals (CNCs). This novel biocompatible and inert nanomaterial is employed as a polymeric matrix to immobilize and stabilize glucose oxidase in the fabrication of a reproducible, operationally stable, highly selective, cost-effective, screen-printed glucose sensor. The sensors have a linear range of 0.1–2 mM (R^2^ = 0.999) and a sensitivity of 5.7 ± 0.3 µA cm^−2^∙mM^−1^. The limit of detection is 0.004 mM, and the limit of quantification is 0.015 mM. The sensor maintains 92.3 % of the initial current response after 30 consecutive measurements in a 1 mM standard glucose solution, and has a shelf life of 1 month while maintaining high selectivity. We demonstrate the practical application of the sensor by monitoring the glucose consumption of a fibroblast cell culture over the course of several days.

## 1. Introduction

Due to the high demand for glucose monitoring for diabetes mellitus management, glucose biosensors have become the most extensively studied biosensing devices [1]. Multiple types of glucose sensors have been developed to provide diagnostic information about various biological samples including blood, urine, interstitial fluid, sweat, breath, saliva, and ocular fluid [2,3,4,5,6,7,8,9,10,11]. In addition, glucose monitoring has been widely applied in the food industry as a quality control and safety tool [12]. Another application is cell culture: Because glucose is a primary carbon source for proliferating mammalian cells, glucose quantification provides valuable information on metabolism [13,14,15]. Methods for glucose monitoring in cell culture include Raman spectroscopy, liquid chromatography, and other spectroscopic techniques; in general, these techniques are expensive and time-consuming [16,17,18,19]. Hence, a convenient and cost-effective alternative to spectroscopic methods would be extremely useful. In this study, an easy-to-produce, low-cost, fast-responding, and reproducible electrochemical enzymatic glucose sensor was developed by combining glucose oxidase and electrochemical mediators on a screen-printed strip format for direct measurement of glucose concentrations in cell culture medium. Cell culture media contain complex mixtures of compound and consequently represent challenging environments for enzymatic glucose sensors. For instance, non-specific binding of proteins and other compounds can interfere with enzyme stability and activity, limiting the sensor’s dynamic range and operational stability as well as other aspects of performance [20,21]. To address these issues, a novel enzyme immobilization matrix material, cellulose nanocrystals (CNCs), was applied to shelter the active enzyme from deactivation while keeping the surface permeable to glucose in a complex medium that still provides good operational stability for reliable and multi-use glucose monitoring in cell culture medium. 

We chose CNCs as the enzyme matrix for this project because they meet the criteria for an ideal carrier matrix for stabilized enzyme immobilization [22]. CNCs are natural and inert polysaccharide-based polymers in which each monomer unit has three hydroxyl groups that stabilize the tridimensional polymer structure by forming inter- and intra-polymer chain hydrogen bonds [23]. The robust structure provides a thermally and chemically stable environment for enzyme immobilization [24], as well as a stiff flat surface onto which the enzyme is immobilized without being entrapped. As a consequence of this arrangement, the minimized steric hindrance allows the enzyme to retain virtually its full activity [25]. Additionally, due to their high aspect ratios and high surface area, nanoscale CNCs are highly efficient for enzyme immobilization [26], and the covalent bonding between tunable CNC functional groups and enzymes should minimize enzyme leakage [27,28]. Last but not least, this matrix provides a biocompatible microenvironment for enzyme immobilization [29]. The use of CNCs as an enzyme support material increases the sensitivity, reproducibility, operational stability, and shelf life of several enzymatic biosensors [30]. 

Despite their promising qualities, the full potential of CNCs as an enzyme scaffold material for the fabrication of electrochemical glucose biosensors has not yet been fully explored. In particular, no previous studies have tested such glucose sensors in complex cell culture media, which are known to decrease sensor performance [31]. To date, no CNC-based sensor has been used for cell culture applications. For example, Esmaeili et al. reported a screen-printed glucose sensor prepared by anchoring glucose oxidase to a CNC/polypyrrole (PPY) layer via physical entrapment [32]. All of the evaluations of this CNC/PPY glucose sensor were conducted in phosphate buffer. In buffer solutions, the sensor exhibited good sensitivity, selectivity, acceptable reproducibility, and shelf life: 95 % of its initial current response was conserved after 17 days of storage at 4 °C under dry conditions. 

Here, we describe for the first time a screen-printed glucose sensor employing 2,2,6,6-tetramethylpiperidine-1-oxyl (TEMPO)-CNC as an enzyme scaffold for stable covalent enzyme immobilization. Carboxylic-CNCs (TEMPO-CNCs) were produced by converting the hydroxymethyl groups of CNCs to their carboxylic form with the TEMPO oxidizing reaction. The carboxyl groups on TEMPO-CNCs were covalently bonded to the glucose oxidase via the carbodiimide coupling method to obtain a stable enzyme immobilization [33] (Figure 1). Sensor performance tests examined operational stability, reproducibility, sensitivity, and selectivity. As a practical use case, the sensors were used to monitor the glucose concentration of the supernatant of a fibroblast cell culture. Performance of the TEMPO-CNC sensor was compared with that of a glucose sensor in which the enzyme was immobilized via a standard method, glutaraldehyde crosslinking [34,35,36]. 

## 2. Materials and Methods

### 2.1. Chemicals and Reagents

Cellulose nanocrystals (8 wt.% CNC gel) were purchased from Blue Goose (Saskatoon, SK Canada). 2,2,6,6-tetramethylpiperidine-1-oxyl (TEMPO, 98%), sodium bromide (NaBr, ≥99.99%), sodium hypochlorite solution (NaClO, 6–14% active chlorine), D-(+)-Glucose, N-(3-Dimethylaminopropyl)-N′-ethylcarbodiimide (EDC, 97%), N-hydroxysuccinimide (NHS, 98%), Glutaraldehyde solution (GA, 25%), glucose oxidase from *Aspergillus niger* (GOx), potassium dihydrogen phosphate (KH_2_PO_4_, ≥99%), sodium chloride (KCl, ≥99%), sodium hydroxide (NaOH, 98% ), hydrochloric acid (HCl, 36.5–38%), bovine serum albumin (BSA, 96%), L-ascorbic acid (AA, A5960), uric acid (UA, ≥99%), L-(+)-lactic acid (LA, L1750), 2-(N-morpholino) methanesulfonic acid (MES, ≥99%), Nafion solution (D2021, 20 wt.%), iron(III) chloride hexahydrate (FeCl_3_·6H_2_O, 98%), potassium ferricyanide(III) ([K_3_Fe(CN)_6_], ≥99.0%), and N, N-dimethylformamide (DMF, ≥99%), were purchased from Sigma-Aldrich (Buchs, Switzerland). Dulbecco’s Modified Eagle Medium (DMEM; 11,966,025 and 11885084) and penicillin–streptomycin (10,000 U/L) (Pen-Strep, 15140122) were purchased from Gibco/Thermo Fisher Scientific (Dreieich, Germany). Standardized fetal bovine serum was purchased from Biochrom (S0615; Merckgroup, Berlin, Germany). Carbon black (CB) of industrial standard grade was obtained from Cabot Corporation (Ravenna, Italy). Milli-Q water was produced using a Merck Milli-Q water purification system.

### 2.2. Apparatus and Screen-Printed Electrodes

Screen-printed electrodes (SPEs) were produced in our laboratory on a DEK 248 screen printing machine (Weymouth, UK). The electrodes consist of a 3 mm diameter graphite-based working electrode (WE), an Ag/AgCl pseudo reference electrode (RE), and a graphite-based counter electrode (CE) printed onto polyester film. The Silver/silver chloride (90/10) ink, polymer dielectric ink, and graphite ink were purchased from the Gwent group (Pontypool, UK) and printed onto a polyester film obtained from MacDermid (San Marcos, CA, USA) (Appendix A).

Electrochemical measurements (cyclic voltammetry, chronoamperometry) were carried out with a PalmSens 4 potentiostat (PalmSens BV, The Netherlands) equipped with the PSTrace software. 

### 2.3. Preparation of CB–Prussian Blue (PB)-Modified SPEs

To improve the electron transfer efficiency of the SPEs, the WE was modified with a carbon black–Prussian blue (CB–PB) mediator layer [37]. In this mediator layer, CB plays a role in the increase of the electroactive area, and PB acts as the mediator for the electron transfer. In general, CB–PB was prepared by adding 1 g CB to 10 mL of 0.1 M [K_3_Fe(CN)_6_] solution (solution in 10 mM HCl); the CB solution was then mixed with a 10 mL preparation of 1 M FeCl_3_·6H_2_O solution (solution in 10 mM HCl). The mixture was stirred for 10 min and centrifuged to remove the supernatant. The precipitate was washed with a 10 mM HCl solution. The washing step was repeated until the supernatant became colorless. The precipitate was collected and dried in an oven at 100 °C for 90 min. The resultant CB–PB powder can be stored in the dark at room temperature for years [38].

For preparation of the CB–PB suspension solution, 1 mg CB–PB powder was dispersed in 1 mL DMF/Milli-Q water (*v*:*v*, 1:1), and the suspension was sonicated in an ultrasonic bath for 90 min at 102 kHz. 

Two microliters of CB–PB suspension were dropped onto the WE and allowed to dry in between every drop at ambient temperature and humidity (repeated three times). The prepared sensors were stored for up to 2 weeks in vacuum packaging at room temperature without light exposure until the next modification step.

### 2.4. Preparation of TEMPO-CNC 

TEMPO-CNCs were produced according to the protocol of Tsuguyuki et al. [39]. Briefly, 12.5 g CNCs (8 wt.% CNC gel), 0.016 g TEMPO oxidant (0.1 mmol), and 0.1 g sodium bromide (1 mmol) were mixed in 100 mL Milli-Q water under magnetic stirring for 30 min to obtain a homogenous CNC suspension. Nine milliliters of NaClO solution (12 %) were adjusted to pH 10 using 0.1 M HCl solution, added dropwise to the CNC suspension to initiate the reaction, and then to a final ratio of 15 mmol NaClO per gram of nanocellulose. The pH value was maintained at pH 10 to complete the oxidation process; to this end, the pH was adjusted by adding 0.5 M NaOH solution dropwise every 30 min during the first 6 h. The reaction mixture was incubated for 18 h at room temperature under magnetic stirring, and the pH was adjusted to 10 using 0.5 M NaOH solution. As a result, the hydroxyl groups of CNCs were oxidized to carboxyl groups, yielding TEMPO-CNC. The resultant reaction mixture was centrifuged at 4500 rpm for 15 min, and the supernatant was discarded. The TEMPO-CNC particles were washed with Milli-Q water until the obtained supernatant reached pH 7. Subsequently, the TEMPO-CNC particles were oven-dried under vacuum at 50 °C for 12 h and stored at room temperature under vacuum packaging.

The TEMPO-CNCs were characterized with Fourier transform infrared spectrometry (FTIR). Absorbance infrared spectra were recorded using a Bruker Vector 2200 Fourier transform infrared (FT-IR) spectrometer equipped with a Goldengate unit using the attenuated total reflectance (ATR) technique. One milligram of dry TEMPO-CNCs and 200 mg crystalline KBr were ground together using an alumina mortar and pestle and pressed to form discs. The discs were scanned over the range 3700 to 800 cm^−1^ with a total of 30 scans at a resolution of ±4 cm^−1^. 1 mg of dry CNCs was characterized with FTIR using the same parameters.

### 2.5. Immobilization of Glucose Oxidase (GOx) onto TEMPO-CNC

Glucose oxidase (GOx) was covalently immobilized onto the TEMPO-CNC by *EDC*/*NHS*-mediated coupling (See Appendix A) [40]. To activate the carboxyl group of TEMPO-CNC, 11 mg dry TEMPO-CNC was added to 1 mL of 0.1 M MES buffer (pH 6.4) containing NHS (13.6 mmol) and EDC (13.6 mmol) and incubated for 3 h at room temperature under mild rotation (Intelli-Mixer: 10 rpm). To covalently bind GOx to TEMPO-CNC, 6 µl GOx solution (5 mg/mL; 0.05 M phosphate buffer, pH 7.4) and 5 µL BSA (2.5 mg/mL; 0.05 M phosphate buffer, pH 7.4) were added to 49 µl of previously preactivated TEMPO-CNC solution. After rotation for 30 min (Intelli-Mixer: 10 rpm) at room temperature, the TEMPO-CNC-GOx reaction solution was incubated at 4 °C overnight to achieve complete covalent bonding and used the next day for sensor modification. TEMPO-CNC-GOx conjugate should be freshly prepared before sensor modification.

### 2.6. Preparation of the TEMPO-CNC Glucose Sensor

The TEMPO-CNC glucose sensor was prepared in a two-step procedure: First, five microliters of TEMPO-CNC-GOx conjugate solution were drop-casted onto the CB–PB-modified WE surface, left in ambient conditions for 30 min to form the enzyme layer, and stored at 4 °C overnight. The surface density of GOx on working electrode is 88.5 ng/mm^2^. The next day, the enzyme layer was covered by dispensing 5 µl of 1 wt.% Nafion solution (in Milli-Q Water). The modified SPE was dried at ambient conditions, vacuum-packed, and stored at 4 °C protected from light exposure until use.

### 2.7. Preparation of the Glutaraldehyde-Based Glucose Sensor (GA Sensor)

The GA sensor was prepared refer to the former research work [41]. 3 µL glutaraldehyde (0.25 % wt.%) was dropped onto the CB–PB-modified WE surface as a crosslinker for the GOx immobilization. Then, 5 µL GOx/BSA solution (2.5 mg/mL GOx and 1 mg/mL BSA in 1 wt.% Nafion solution) were deposited on the WE. The modified SPE was dried at ambient conditions and subsequently stored under the same conditions as for the TEMPO-CNC glucose sensor.

### 2.8. Characterization of the Modification Step of the TEMPO-CNC Glucose Sensor

SEM images of different modification layers on the SPE were acquired using the Hitachi S-3400N Scanning Electron Microscope (SEM) at accelerating voltages of 20 kV and working distances of 14.3 mm. The samples were coated with a layer of gold (~5 nm) using a Cressington 108auto Automatic Sputter Coater.

The sensor was tested by cyclic voltammetry before and after each modification step to validate the presence of the CB–PB modification layer and the electron transfer efficacy of the TEMPO-CNC layer. In the measurement, 100 µL phosphate buffer (0.05 M KH_2_PO_4_ and 0.1 M KCl at pH 7.4.) for electrochemical measurement were added to the SPEs, and potential was applied with a range of −0.2 to +0.5 V and a scan rate of 50 mV·s^−1^ versus the RE. All electrochemical experiments were carried out with a PalmSens 4 potentiostat (PalmSens BV, Houten, the Netherlands). The current profile was recorded and used for the analysis. 

### 2.9. Calibration of the TEMPO-CNC Glucose Sensor in Buffer Solution

The sensitivity, linear range response, limit of detection, and limit of quantification of the developed TEMPO-CNC glucose sensor were characterized through amperometric measurement. In the presence of glucose and oxygen, the enzymatic reaction catalyzed by glucose oxidase produces H_2_O_2_, which oxidizes PB from the reduced (PB_red_) to the oxidized state (PB_ox_). The formed PB_ox_ is reduced back to PB_red_ on the WE at −50 mV (versus RE). The current generated by the reduction of PB_ox_ is proportional to the glucose concentration [42].

During the measurement, phosphate buffer without glucose, and phosphate buffer supplemented with glucose concentrations from 0.1 to 5.0 mM, were added to the SPE. The concentration range was chosen based on the glucose concentration of low-glucose DMEM culture medium (5.5 mM) used for cell culture.

The generated current response was recorded by applying a −50 mV potential between the RE and WE on the SPEs for 30 s. An average current value was achieved by measuring each glucose concentration three times, including a wash step using the same buffer between each measurement. The background current was measured in the same way but in the absence of glucose, and the background value was subtracted from each measurement. The average current value at each glucose concentration was plotted against glucose concentration to obtain a calibration curve. The limit of detection (LOD) and limit of quantification (LOQ) were calculated based on the formulas LOD= 3S/b and LOQ=10 S/b**,** where S is the standard deviation of the background measurement (n = 3), and b is the slope of the calibration curve [43]. The intra-batch reproducibility of the TEMPO-CNC glucose sensor was evaluated by calculating the RSD % of the slopes from the obtained calibration curves from five different sensors. All measurements were also carried out with the GA sensor fabricated in our work.

### 2.10. Selectivity Testing of the TEMPO-CNC Glucose Sensor

In the selectivity assessment of the TEMPO-CNC glucose sensor, L-ascorbic acid (AA 0.1 mM), uric acid (UA; 0.1 and 1 mM), and L-(+)-lactic acid (lactate; 0.1 and 1 mM) were chosen as the interfering species because they are also present in cell culture media [44,45,46] and could therefore interfere with the glucose measurement. Through amperometric measurement (as described in paragraph 2.9), the current response of the TEMPO-CNC glucose sensor was compared between 1 mM glucose standard solution and the same solution spiked with each interfering species. The relative current response was used to evaluate the interference effect by the various species. The relative interference response was calculated as ΔI/I×100 %, in which ΔI is the current difference between measurements in the 1 mM glucose standard solution with or without interfering species, and I is the current measurement in 1 mM glucose standard solution without interfering species. The lower the relative interference response, the higher the selectivity of the sensor toward glucose. The selectivity of the prepared GA sensor was measured using the same method.

### 2.11. Operational Stability of the TEMPO-CNC Glucose Sensor

The operational stability measurement of the TEMPO-CNC glucose sensor was carried out by running 30 consecutive amperometric measurements in 1 mM glucose solution in phosphate buffer. After each test, the SPE was washed and measured with phosphate buffer without the addition of glucose to measure the baseline, which did not show any drift over the course of all measurements. The ratio of the current response registered in the final measurement to the current obtained from the initial measurement was used to evaluate the operational stability. Additionally, the standard deviation and RSD % were calculated for all recorded current responses from 30 different measurements. The operational stability test was conducted identically for the GA sensor.

### 2.12. Shelf life of the TEMPO-CNC Glucose Sensor

For sensor shelf-life measurements, a batch of 50 TEMPO-CNC glucose sensors were prepared. Of these sensors, three groups of 15 sensors were stored under one of the following conditions: (**a**) 400 mbar nitrogen vacuum packaging; (**b**) vacuum package (≤50 mbar); and (**c**) after high-vacuum drying treatment (<10 mbar) in a vacuum chamber for complete drying, followed by storage in a vacuum package (≤50 mbar). All sensors were stored at 4 °C for 1 week, 2 weeks, 1 month, or 2 months. After storage, the sensors were calibrated in glucose standard solutions in phosphate buffer. A sensor’s shelf life was calculated using the expression S/Sint×100 %, where S is the sensitivity of a sensor after storage, and Sint is the sensitivity of a sensor from the same batch before storage. All measurements were carried out in triplicate. The shelf life of the GA sensor was evaluated using the same protocol.

### 2.13. NIH 3T3 Cell Culture 

NIH 3T3 fibroblasts were cultured in DMEM medium (5.5 mM glucose) supplemented with 10% FBS and 1% penicillin–streptomycin at 37 °C and 5% CO_2_ in a humidified atmosphere. Cells were cultured in 75 cm^2^ culture flasks with a total medium volume of 15 mL and kept below 70% confluence.

For sensor measurements, cells were left to proliferate in the cell culture flask for 4 days without passage or changing the cell culture medium. Every day, 1 mL culture supernatant was collected and immediately frozen at −20 °C. The sample taken at day 0 was centrifuged for 5 min at 200 g to prevent carry-over of suspended cells.

### 2.14. Investigation of the Matrix Effect of Cell Culture Medium

The presence of complex components in cell culture medium, such as growth factors, serum, and antibiotics, can interfere with the electrode and might affect sensor sensitivity. The sensor was initially calibrated with standard glucose solutions (0.1–2 mM) in both phosphate buffer and DMEM medium. To minimize the matrix effect of complete DMEM medium on sensor performance, the medium was diluted 1:4 (*v*:*v*) in phosphate buffer, spiked with different concentrations of glucose (0.1–2 mM), and used to obtain a calibration curve. The medium used in these measurements was the same DMEM medium as was used for cell culture but without glucose. The relative percentage of sensor current response is used to evaluate the matrix effect on sensor sensitivity with the following formula:

The relative percentage of response = Sm/Sb×100 %, where Sm is the sensitivity of the glucose sensor measured in standard glucose solution in the different matrix, and Sb is the sensitivity of the glucose sensor measured in standard glucose solution in phosphate buffer.

### 2.15. Glucose Concentration Monitoring in Fibroblast Cultures

Frozen fibroblast cell culture supernatant acquired on days 0–4 was thawed, brought to room temperature, and diluted at a ratio of 1:4 (*v*:*v*) in phosphate buffer. The current of the cell culture supernatant was measured via the amperometric method with the calibrated TEMPO-CNC glucose sensor. The glucose concentration was calculated based on the calibration curve derived in completed DMEM medium diluted 1:4 (*v*:*v*) in phosphate buffer.

## 3. Results and Discussion 

### 3.1. Characterization of TEMPO-CNCs with Fourier Transform Infrared Spectrometry (FTIR)

The TEMPO-CNCs were synthesized by converting the hydroxymethyl groups of the CNCs to their carboxylic form with the TEMPO oxidant. To confirm the TEMPO oxidation reaction, FTIR spectra of TEMPO-CNCs and unmodified CNCs were recorded (Figure 2). Notably, in comparison with pristine CNCs, TEMPO-CNCs showed a more intense peak at 1607 cm^−1^, which is attributed to the C**=**O stretching of the carbonyl of the carboxylic acid groups [47]. 

### 3.2. Characterization of the Sensor Functionalization

The morphology of the different layers coated on the electrode was characterized by SEM. The unmodified bare SPE with carbon paste exhibited a homogeneous morphology (Figure 3A). Figure 3B shows the image of the surface of the electrode modified by CB–PB, revealing a rougher morphology of the electrode surface compared with the bare SPE, which can be attributed to CB particles (around 10 µm) as reported previously [48]. The TEMPO-CNC-GOx modified layer (Figure 3C) exhibited a three-dimensional nanostructure with a typical cylinder shape that densely and homogeneously covered the SPE. TEMPO-CNCs formed a uniform and porous structure that offered a large surface area well suited for highly efficient loading of the GOx. The GOx coating was well distributed, with no phase separation, and could therefore ensure efficient electron transfer from the modification layer to the CB–PB mediator layer below. The Nafion layer fully and homogenously covered the SPE (Figure 3D).

Cyclic voltammetry (CV) is used to characterize three types of WEs: (a) bare, (b) after CB–PB modification, and (c) after TEMPO-CNC-GOx and Nafion modification. Detection of the typical oxidation and reduction peak of CB–PB was used to validate the successful modification of the CB–PB layer. It also determined the optimal electron transfer efficiency of the mediating layer and, hence, the highest possible electrode transfer efficiency of the final sensor. The CV profile (Figure 3E) of the electrode after TEMPO-CNC-GOx and Nafion modification provides information about whether the inert nanocellulose-based layer and other added materials decreased the electron transfer efficiency between the mediator (PB) and electrode surface. As expected, the bare electrode did not exhibit any typical redox peaks. By contrast, after modification with CB–PB, the electrode exhibited the typical oxidation and reduction peaks, which are both about 0.1 V given by the oxidization and reduction processes of PB [49]. After modifying the CB–PB functionalized electrodes with the TEMPO-CNC-GOx and the Nafion layers, the CV profile was very similar to that obtained with the SPE modified with CB–PB alone. The homogeneous distribution of the TEMPO-CNC-based layer on the modified electrode surface also contributed to the almost overlapping CV profiles. The only difference was the slightly lower current of the oxidation peak, which could be due to the non-conductive TEMPO-CNC-GOx and Nafion layers, which slightly decrease electron transfer between the mediator layer (PB) and the electrode surface. However, this effect was very small and could therefore be neglected.

### 3.3. Calibration of the TEMPO-CNC Glucose Sensor in Buffer Solution

To investigate the sensor performance, we calibrated the prepared TEMPO-CNC glucose sensor in standard glucose solution in phosphate buffer (Figure 4). The sensor was linear between 0.1 and 2.0 mM glucose (y = 0.40x − 0.018, R^2^ = 0.99) with a sensitivity of 5.7 ± 0.3 µA∙cm^−2^ ∙mM^−1^ (n = 5 sensors). The LOD and LOQ were 0.004 and 0.015 mM glucose, respectively (*cf.*
Appendix A). The intra-batch reproducibility of the sensors, evaluated by calculating the RSD % value of the sensitivity of five TEMPO-CNC glucose sensors, was 4.6 %. All results were compared with those obtained with the GA sensor prepared in this study (Appendix A). The GA sensor had a similar LOD (0.004 mM) and LOQ (0.012 mM) with a similar linear range (0.1–2.0 mM) (Appendix A); however, the RSD was 10.2 % for five sensors, much less reproducible than the TEMPO-CNC glucose sensor.

### 3.4. Selectivity Test of the TEMPO-CNC Glucose Sensor

Although enzymes convert their substrates with high selectivity, the enzymatic conversion of a sensor can still be influenced by other compounds present in the solution to be analyzed. Uric acid (UA), L-lactic acid (LA), and L-ascorbic acid (AA) are metabolites formed in cell cultures that can interfere with the enzymatic glucose sensor measurement [44,45,46]. The physiological concentration is below 0.1 mM for AA and ranges from 0.24 to 0.35 mM for UA and 0.3 to 1.3 mM for LA.

We evaluated the TEMPO-CNC glucose sensor’s selectivity by comparing the current measured in a 1 mM glucose standard solution with the current measured in a 1 mM glucose standard solution spiked with 0.1 mM AA, 0.1 mM UA, 1 mM UA, 0.1 mM LA, or 1 mM LA. The TEMPO-CNC sensor gave a low relative response for each interfering species (Figure 5). The highest relative response was recorded for 1 mM LA (7.2%). All other spiked compounds induced relatively small responses (<5%). Notably, the relative responses induced by the individual compounds did not strongly correlate with their spiked concentrations. It should be noted that the TEMPO-CNC glucose sensor had better selectivity than the GA sensor. The GA sensor was not pre-selective enough to eliminate the interfering effect from the spiked species in solution (Appendix A). Relative responses to interfering species were much higher, especially for AA (30 % at 0.1 mM) and LA (30 % at 1 mM). The superior selectivity of the TEMPO-CNC sensor could be attributed to a barrier effect offered by the mesoporous structure of the TEMPO-CNC membrane and the separate coating layer of Nafion, both of which help to eliminate the interfering species from the surface of the electrode. 

### 3.5. Operational Stability of the TEMPO-CNC Glucose Sensor

To further explore the TEMPO-CNC glucose sensor as a stable operational tool for repetitive glucose measurements in cell cultures, a single sensor was used for 30 individual, repeated amperometric measurements in a 1 mM glucose solution. Figure 6 shows the average values, with standard deviation, from measurements acquired using three individual sensors. The ratio of the current response of the last measurement to the initial measurement was used to evaluate the operational stability. The TEMPO-CNC glucose sensor retained about 92.3% of the initial current response after 30 measurement cycles; the standard deviation of the current response in 30 separate measurements was ±3.7, and RSD % was 3.9%. The results show that the TEMPO-CNC glucose sensor offers better operational stability than the GA sensor (85.3% of the initial current retained after 30 measurements: Appendix A). Ninety percent of original activity is a reasonable threshold for expiration of the sensor; hence these results confirm that the TEMPO-CNC glucose sensor has satisfactory operational stability and can be applied to multi-cycle measurements [50].

### 3.6. Shelf Life of the TEMPO-CNC Glucose Sensor

To investigate the shelf life and optimal storage conditions, 50 TEMPO-CNC glucose sensors (single batch) were prepared and stored for various time periods under different conditions. The sensors were stored at 4 °C under three conditions: (a) 400 mbar nitrogen vacuum packaging; (b) vacuum package (≤50 mbar); and (c) a combination of high-vacuum drying treatment at <10 mbar followed by vacuum packaging (≤50 mbar). All sensors were tested for their remaining activities after 1 week, 2 weeks, 1 month, and 2 months of storage under the specified conditions. The shelf lives of the TEMPO-CNC glucose sensors were evaluated according to their remaining activities after storage, and the activities were normalized against the initial activity, i.e., without storage. High-vacuum drying pretreatment followed by storage under vacuum provided the best shelf life: approximately 90 % and 60 % of the initial activity of the TEMPO-CNC glucose sensor was retained after 1 and 2 months, respectively (see Figure 7). By contrast, after 2 months of storage in the vacuum package without high-vacuum drying treatment, the sensors had such low activity that no calibration curve could be obtained. The very simple solution of storage under a low-pressure nitrogen atmosphere preserved the full initial activity for 2 weeks, after which the remaining sensor activities dropped to ~50 %.

We also evaluated the shelf life of the GA sensor. The prepared sensor kept 70 % of its initial activity under 400 mbar nitrogen vacuum packaging; however, after 1 month of storage, the sensor lost more than 80% of its activity (Appendix A). Storage in a vacuum package with high-vacuum drying treatment did not improve the shelf life to the same extent as for the TEMPO-CNC sensor, with activity dropping to 50% after 2 weeks.

These results demonstrated that the TEMPO-CNC glucose sensor has a longer shelf life than the GA sensor. Thus, as hypothesized in the literature [27], the use of TEMPO-CNC as an enzyme scaffold provides an enzyme-stabilizing microenvironment and thus extends the shelf life. In addition, by optimizing the storage conditions for enzymatic sensors in general, the shelf lives of both sensors could be improved.

### 3.7. Comparing the Performance of the TEMPO-CNC Glucose Sensor with Those of Previously Reported Sensors

Next, we compared the linear response range, the LOD, and the shelf life of the TEMPO-CNC-based glucose sensor with those of enzymatic (glucose oxidase) glucose sensors reported previously. The sensors listed in Table 1 were modified with various materials (metal oxide: ZnO/graphene-carbon nanotube (GR–CNT–ZnO) [51]; conductive polymers: Polypyrrole (PPY) [32]; polyaniline (PANI)/PB [52]; PEDOT:PSS [53] and cellulose-based material, i.e., paper) [54]. The sensors were produced using either SPE or glassy carbon electrode (GCE) technology. Among the SPE-based sensors, the TEMPO-CNC glucose sensor offered similar performances in terms of linear range and LOD [32,52,53,54]. However, it had one of the longest shelf lives in its class and was outperformed only by sensors consisting of more elaborate and expensive materials, such as the GCE [51].

### 3.8. Investigation of the Matrix Effect of Cell Culture Medium

The TEMPO-CNC glucose sensor was designed for use in cell culture glucose monitoring. Cell culture medium contains many complex components, such as serum and antibiotics, that could exert interference and thus decrease the sensor sensitivity. To investigate the matrix effect on the sensor sensitivity and to identify ways to diminish these effects, we carried out the following experiments. 

Figure 8 shows the comparison of the glucose calibration curves sequentially recorded in different matrixes. For all cases, a linear range from 0.1 to 2 mM could be confirmed for the TEMPO-CNC glucose sensor. However, the sensitivity decreased from 6.2 µA∙cm^−2^∙mM^−1^ in phosphate buffer to 2.9 µA∙cm^−2^∙mM^−1^ (45.8 %) in complete cell culture medium. The matrix effect was mitigated by diluting the cell culture medium 1:4 in phosphate buffer, after which the sensor maintained a sensitivity of 5.9 µA∙cm^−2^∙mM^−1^, i.e., a 95.2 % relative response. The positive effect of this very simple strategy enabled glucose quantification in samples containing concentrations of up to 8 mM.

### 3.9. Glucose Concentration Monitoring in Fibroblast Cell Culture

Finally, we monitored glucose consumption of a NIH 3T3 fibroblast culture by quantifying the glucose concentration in the cell culture supernatant collected at different time points from the same culture flask (days 0, 1, 2, 3, and 4). To diminish the matrix effect, the medium was diluted at a ratio of 1:4 (*v*:*v*) with phosphate buffer prior to the measurements. The measured glucose concentration was interpolated using a calibration curve generated with standard glucose calibrant medium (0.1 to 2.0 mM) in glucose-free cell culture medium diluted 1:4 (*v*/*v*) with phosphate buffer. The results shown in Figure 9 show that glucose concentration decreased as a result of glucose consumption during the cell culture period. 

## 4. Conclusions

We developed a low-cost, simple, and operationally-stable glucose sensor employing TEMPO-CNC as a scaffold for a stable immobilization of glucose oxidase. In addition, this is the first study to use glucose oxidase functionalization on TEMPO-CNC for making screen-printed electrochemical glucose sensors. Using our new sensor, we were able to successfully monitor the glucose consumption of the cell culture. Relative to an electrochemical glucose sensor with standard enzyme immobilization based on chemical crosslinking, the TEMPO-CNC glucose sensor exhibited better operational stability in repetitive tests, higher selectivity, and longer shelf life. However, the linear range of our developed glucose sensor is limited to the range between 0.1 mM to 2 mM, which merits future improvement. For the moment, a dilution of the culture medium must be done before the test in case the glucose concentration in the cell culture exceeds 2 mM. In the future, due to its improved stability in repetitive tests, our sensor has the potential to be combined with microfluidic devices for automated glucose monitoring in cell cultures, which would be of great interest for applications such as organ-on-chip systems and tissue engineering. Because nanocellulose is a wood-derived polymer, it also represents an ideal candidate for fabrication of green electronics and the next generation of disposable point-of-care devices [30]. Additionally, as a matrix for immobilizing different sorts of bio-recognition elements, such as enzymes, antibodies, and DNA sequences, nanocellulose could be applied in developing disposable sensors for food safety and environmental monitoring.

## Figures and Tables

**Figure 1 biosensors-10-00125-f001:**
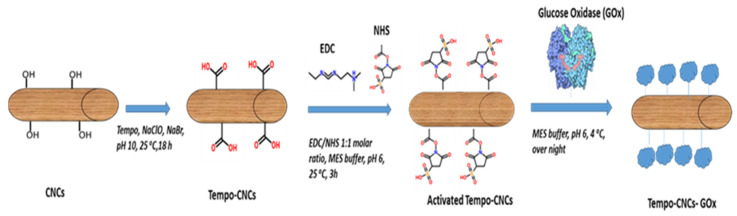
Schematic view of the TEMPO-CNCs oxidation and enzyme immobilization process.

**Figure 2 biosensors-10-00125-f002:**
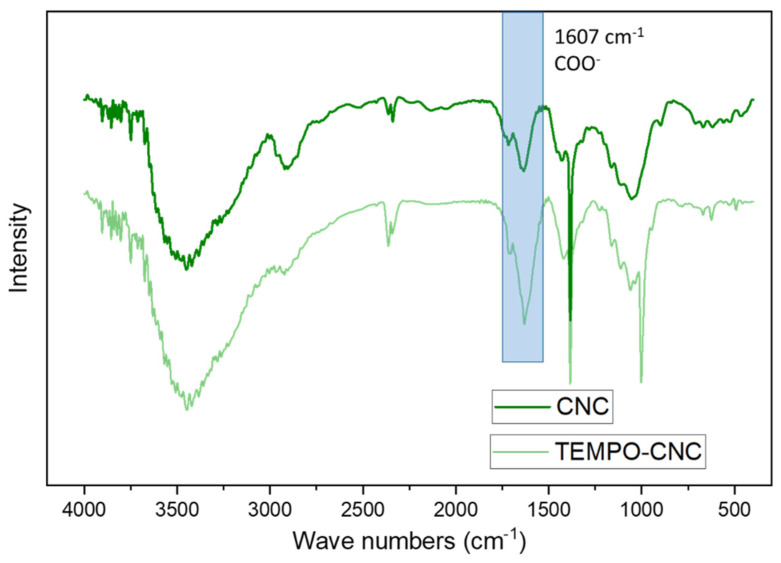
FTIR spectra of cellulose nanocrystals (CNCs) (dark green curve), TEMPO CNCs (light green curve curve).

**Figure 3 biosensors-10-00125-f003:**
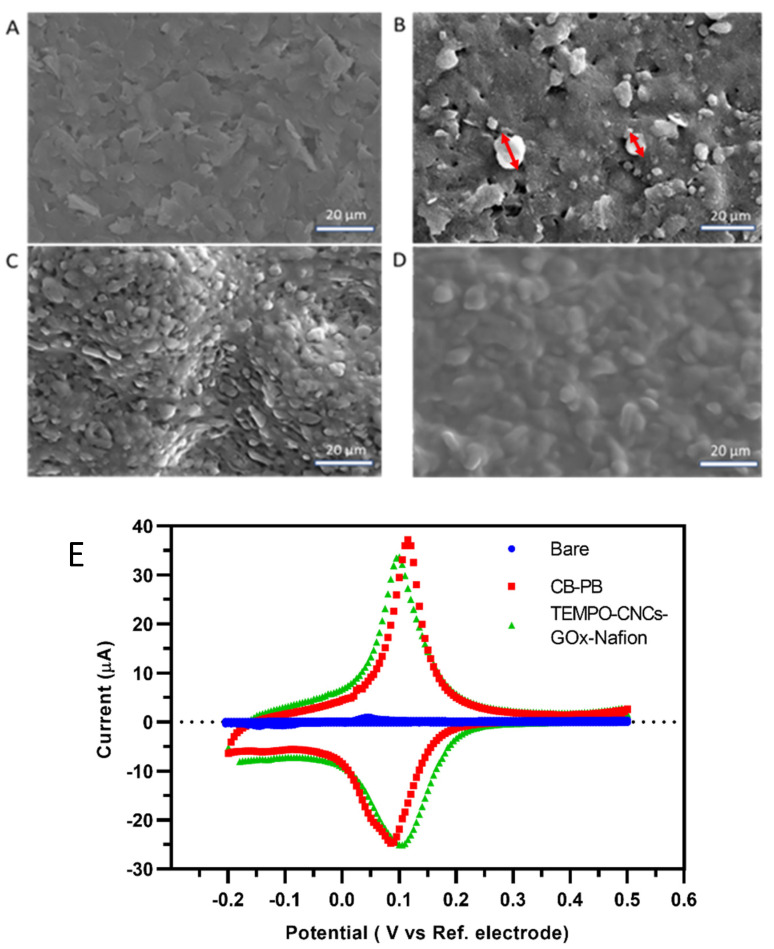
(**A**) SEM image of the bare SPE. (**B**) SEM image of the carbon black–Prussian blue (CB–PB) layer. White arrows exemplarily mark CB microparticles and white triangles exemplarily mark PB nanoparticles. (**C**) SEM image of the TEMPO-CNC-GOx layer. (**D**) SEM image of the Nafion layer. (**E**) Cyclic voltammetry of the TEMPO-CNC glucose sensor after different modification steps: bare (blue dots), after CB–PB modification (red squares), and after TEMPO-CNC-GOx and Nafion modification (green triangles) in phosphate buffer.

**Figure 4 biosensors-10-00125-f004:**
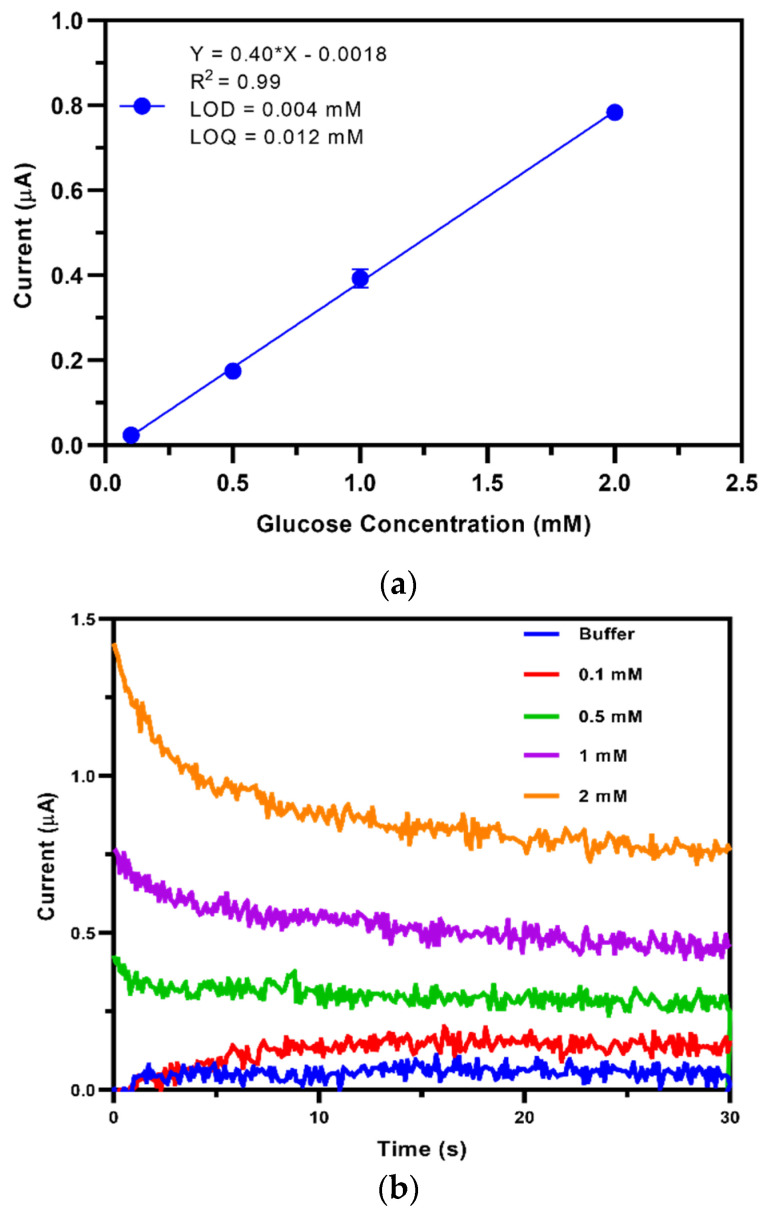
(**a**) Raw data sets of chronoamperometry measurements with the TEMPO-CNC glucose sensor in phosphate buffer containing 0.1–2 mM glucose. (**b**) Calibration curve of the TEMPO-CNC glucose sensor in 0.1–2 mM glucose in phosphate buffer. The linear range (up to 2 mM) was smaller than the entire dynamic range. n = 3; plotted are the mean ± standard deviation and the linear fit from 0.1 to 2 mM.

**Figure 5 biosensors-10-00125-f005:**
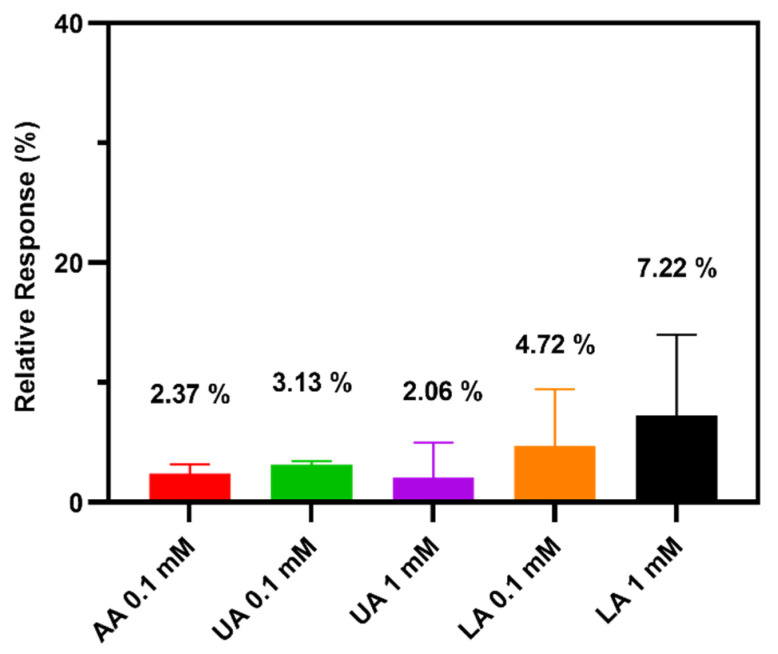
Relative responses of the TEMPO-CNC glucose sensor toward interfering species (0.1 mM L-ascorbic acid, 0.1 and 1 mM uric acid, 0.1 and 1 mM L-lactic acid) in a 1 mM glucose standard solution in phosphate buffer. n = 3 for each condition; 1 mM glucose without interfering species is defined as 100 % response.

**Figure 6 biosensors-10-00125-f006:**
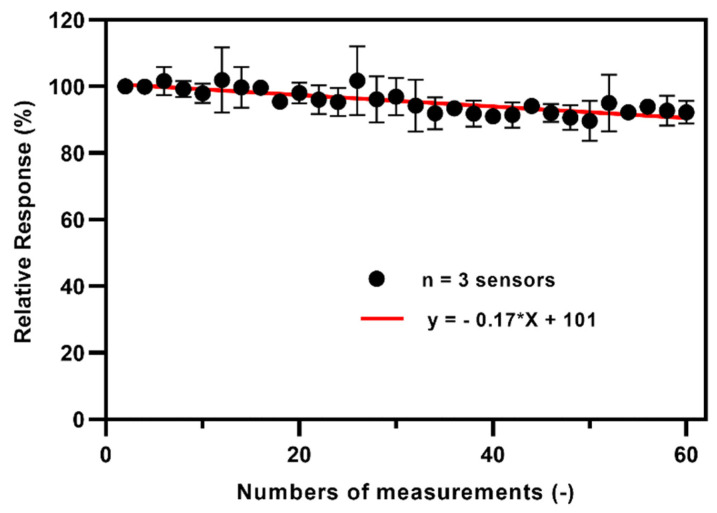
Operational stability of the TEMPO-CNC glucose sensors over 30 repeated measurements in 1 mM glucose solution in phosphate buffer. The graph depicts the average values of three individual sensors (mean ± standard deviation); the red linear fit indicates the loss in measured current over the 30 cycles (total of 6 h of measurement). The average remaining activity of the TEMPO-CNC glucose sensor after 30 measurements was 92.3% of the initial current.

**Figure 7 biosensors-10-00125-f007:**
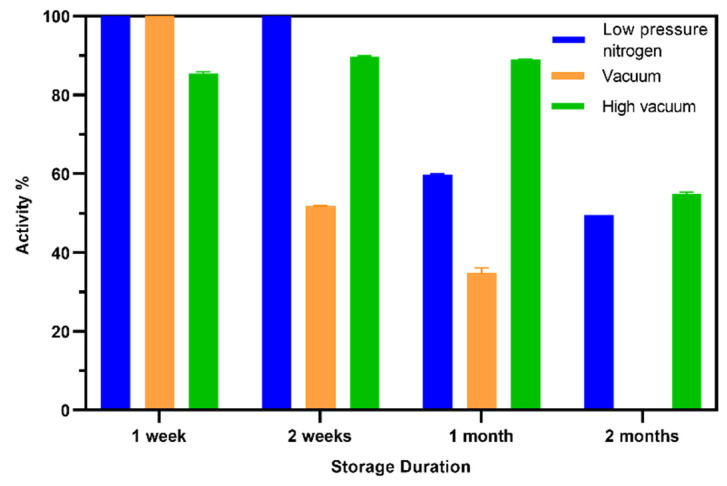
Shelf-life test of the TEMPO-CNC glucose sensor under different storage conditions: low-pressure nitrogen (400 mbar N_2_), vacuum (≤50 mbar), and high-vacuum (≤10 mbar) drying treatment before vacuum packing (≤50 mbar). All sensors were stored at 4 °C. The sensors were tested after 1 week, 2 weeks, 1 month, and 2 months. All values are normalized against the signals of sensors tested without storage.

**Figure 8 biosensors-10-00125-f008:**
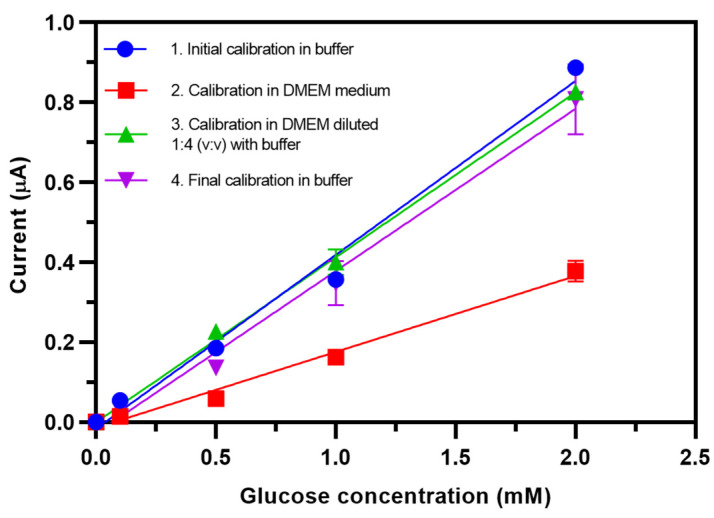
Calibration curve of a single TEMPO-CNC glucose sensor in the range from 0.1 to 2.0 mM glucose in the following order: (1) Phosphate buffer, (2) DMEM medium, (3) diluted DMEM medium (1:4 [*v*:*v*] with phosphate buffer), and (4) phosphate buffer. DMEM refers to DMEM medium supplemented with 10 % serum and 1 % antibiotics. All data points represent mean values ± standard deviation of three measurements.

**Figure 9 biosensors-10-00125-f009:**
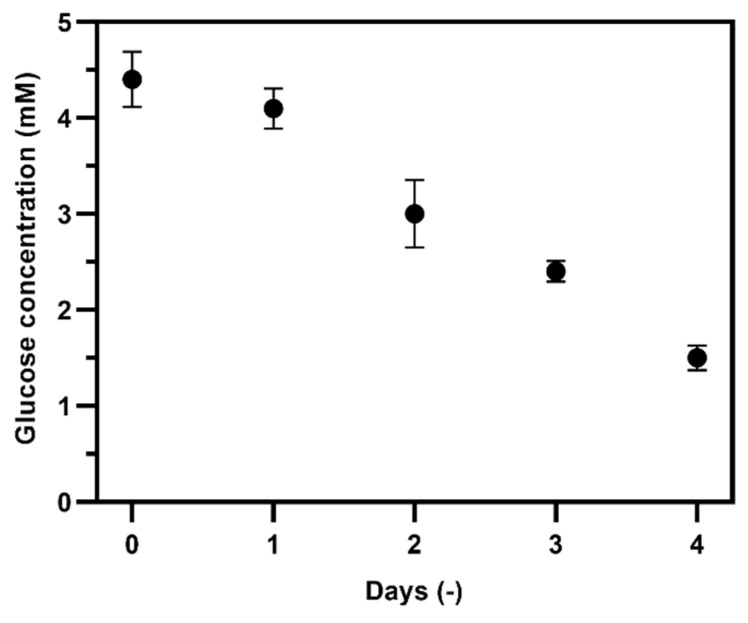
Glucose concentration in fibroblast cell culture supernatant sampled at different time points (day 0 to day 4, n = 3), determined using the TEMPO-CNC glucose sensor.

**Table 1 biosensors-10-00125-t001:** Comparison of the TEMPO-CNC glucose sensor with previously reported enzymatic glucose biosensors. SPE: screen-printed electrode; DPV: differential pulse voltammetry; GCE: glassy carbon electrode; IPE: Injekt printing electrode; N/A: not available.

Modification of Electrode	Detection Method	Linear Range [mM]	LOD [mM]	Shelf Life (Remaining Activity)	Storage Conditions	Reference
TEMPO-CNC (SPE)	Amperometric	0.1–2.0	0.004	90 % after (2 weeks) 90 % (1 month) 60 % (2 months)	4 °C under dry high-vacuum conditions	this work
GA sensor (SPE)	Amperometric	0.1–2.0	0.004	70 % (2 weeks) <10 % (1 month) <10 % (2 months)	4 °C under low-pressure nitrogen	this work
GR–CNT–ZnO (GCE)	Amperometric	0.01–6.5	4.5 × 10^−3^	94.6 % (1 month)	4 °C	[51]
CNCs/PPY (SPE)	DPV	1.0–20	0.05	95 % (2.4 weeks)	4 °C under dry high vacuum conditions	[32]
PANI/PB (SPE)	Amperometric	0.002–1.6	0.4 × 10^−3^	95 % (2 weeks)	4 °C	[52]
PEDOT:PSS (IPE)	Amperometric	0.02-1.0	N/A	80 % (1 month)	4 °C	[53]
Paper disk (SPE)	Amperometric	0.25–2.0	N/A	72 % (1.5 months)	4 °C	[54]

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
