# Peer review of "Screen-Printed Glucose Sensors Modified with Cellulose Nanocrystals (CNCs) for Cell Culture Monitoring"

_biosensors, 2020, doi:10.3390/bios10090125_

Round 1

Reviewer 1 Report

In this manuscript, the authors developed a low-cost, and reproducible sensor based on a cellulose-based material, 2,2,6,6-tetramethylpiperidine-1-oxyl (TEMPO) oxidized-cellulose nanocrystals (CNCs) for monitoring the glucose concentration in cell culture medium. The sensor has a linear range of 0.1 to 2 mM with a limit of detection 0.004 mM. The sensor exhibits good stability and selectivity. Specific comments,

  1. The crystal nature of CNCs should be fully characterized by XRD or other method.
  2. The surface density of immobilized glucose oxidase (GOx) on electrode surface should be estimated.

Reviewer 2 Report

Please find attached my comments

Reviewer 3 Report

The paper is very interesting and sufficiently readable, on the other hand the following revisions are needed in each section as follows:

1) The introduction is clear and all the objectives well stated. At the same time, some recent technology developments are missing such as:

_ metamaterials [Inverse-designed metastructures that solve equations, Science 363 (6433), 1333-1338, 2019]
_ near-zero-index materials [Near-zero-index wires, Optics express 25 (20), 23699-23708, 2017]
_ plasmonics [Plasmonic Optical and Chiroptical Response of Self-Assembled Au Nanorod Equilateral Trimers, ACS nano, 2019]
_ metasurface [Curvilinear MetaSurfaces for Surface Wave Manipulation, Scientific reports 9 (1), 3107, 2019]
_ graphene [Graphene acoustic plasmon resonator for ultrasensitive infrared spectroscopy, Nature Nanotechnology 14, 313–319, 2019]
_ nanoparticles [Electromagnetic and thermal nanostructures: from waves to circuits, Engineering Research Express 2 (1), 015045, 2020]
_ AI structures [Artificial Intelligence and COVID-19: Deep Learning Approaches for Diagnosis and Treatment, IEEE Access 8, 109581-109595, 2020]

It would be beneficial for the reader if authors include such technologies in the introduction section to have a complete picture of the state-of-art.

2) To explore the device behaviour and analyse its sensitivity , authors can consider the following interesting electromagnetic phenomena:
_ surface waves [Modeling and design for electromagnetic surface wave devices, Radio Science 52 (9), 1049-1057, 2017]
Include such phenomena in your model and explain how they can affect the device properties.

3) The paper lack in application examples. Besides sensing and diagnostics, consider wider application fields such as  telecommunications, absorbers, measurements, nanoelectronics, automotive.
I would suggest to comment more on such potential applications by explaining how you can use your device for them.
Please highlight what's new in yours.

4) No limitations of the proposed method have been highlighted.

5) No future improvements/works have been discussed.

Author Response

Please find the authors' response in the attachment.

Reviewer 4 Report

The manuscript entitled “Screen-printed glucose sensors modified with cellulose nanocrystals (CNCs) for cell culture monitoring” describes a new glucose sensor employing TEMPO-CNC as a scaffold for a stable immobilization of glucose oxidase. The innovative results deal with the use TEMPO-CNC in the fabrication of glucose sensors with an electrochemical transduction. For this reason, TEMPO-CNC should be well described in the manuscript both in the introduction and result. Since the topic is interesting, the manuscript can be published after addressing the following comments:

Major comments:

  • The literature concerning cellulose nanocrystals oxidation with TEMPO should be discussed in the introduction. More details about TEMPO-CNC should be reported in the result section. Figure S2 should be moved into the manuscript. How did authors confirm the cellulose modification? What are the chemical modification of cellulose after tempo oxidation? More evidences should be discussed in the text.
  • The authors compared the results of TEMPO-CNC sensor with a so-called standard glucose electrode. The name “standard glucose sensor” recalls a commercial device, but it is prepared by the authors. standard glucose sensor is misleading and it should be changed. During the preparation of TEMPO-CNC and standard glucose sensor, the NAFION membrane is deposited in different ways. What is the effect on sensor performances?
  • Calibration plot (figure 3). Raw data (I vs time curves recorded at different glucose concentrations) should be reported in the manuscript
  • Have authors compared the glucose concentrations determined with the TEMPO-CNC sensor in the culture medium with the glucose concentrations obtained with a consolidated procedure?

Minor comments:

Figure 1 B. The arrows are difficult to see.

Row 207: Was the glucose solution added to SPE?

What is the typical range of the glucose concentration in cell culture medium?

Author Response

(The authors gave the same response as above.)

Round 2

Reviewer 4 Report

The manuscript can be pubblished in the present form